# Pheromone Activity after Stimulation with Ampicillin in a Plasmid-Free *Enterococcus faecalis* Strain

**DOI:** 10.3390/microorganisms10112294

**Published:** 2022-11-19

**Authors:** José Arellano-Galindo, Sergio Zavala-Vega, Rosario Vázquez-Larios, Sara A. Ochoa, Ariadnna Cruz-Córdova, Adolfo Sierra-Santoyo, Lourdes López-González, Rigoberto Hernández-Castro, Silvia Giono-Cerezo, Juan Xicohtencatl-Cortes

**Affiliations:** 1Laboratorio de Bacteriología Médica, Departamento de Microbiología, Escuela Nacional de Ciencias Biológicas, Instituto Politécnico Nacional, Ciudad de México 11340, Mexico; 2Programa de Doctorado en Ciencias Químico Biológicas, Instituto Politécnico Nacional, Ciudad de México 11340, Mexico; 3Laboratorio de Virologìa Clínica y Experimental, Unidad de Investigación en Enfermedades Infecciosas, Hospital Infantil de México Federico Gómez, Ciudad de México 06720, Mexico; 4Laboratorio de Neuropatología, Instituto Nacional de Neurología y Neurocirugía Manuel Velasco Suárez, Ciudad de México 14269, Mexico; 5Departamento de Microbiología, Instituto Nacional de Cardiología Ignacio Chavez, Ciudad de México 14080, Mexico; 6Laboratorio de Investigación en Bacteriología Intestinal, Hospital Infantil de México Federico Gómez, Ciudad de México 06720, Mexico; 7Laboratorio de Toxicología de Plaguicidas y Disrupción Endócrina, CINVESTAV IPN, Ciudad de México 07360, Mexico; 8Departamento de Ecología de Agentes Patógenos, Hospital General Dr. Manuel Gea González, Ciudad de México 14080, Mexico

**Keywords:** pheromones, aggregation substances, ampicillin, antimicrobial resistance, plasmid

## Abstract

Enterococci exhibit clumping under the selective pressure of antibiotics. The aim of this study was to analyze the effect of supernatants from a plasmid-free clone (C29) of *Enterococcus faecalis* subjected to 0.25×, 0.5×, and 0.75× of the minimal inhibitory concentration (MIC) of ampicillin on the expression of an aggregation substance (AS) by a donor plasmid clone (1390R). A clumping assay was performed. The relative expression of *prgB* (gene that encodes AS) was determined and semiquantified in 1390R, and *iad1* expression was determined and semiquantified in C29. AS expression was analyzed in the stimulated 1390R cells by confocal microscopy, flow cytometry, and ELISA. Adherence was also measured. Maximal clumping was observed with the pheromone medium 0.25×. Only the 1390R strain stimulated with the C29 supernatant without ampicillin and with 0.25× was able to express *prgB*. No expression of *prgB* was observed at 0.5× and 0.75×. The difference in relative expression (RE) of 1390R without ampicillin and with 0.25× was 0.5-fold. AS expression in 1390R showed the greatest increase upon stimulation with 0.25×. When 1390R was stimulated with 0.5× and 0.75×, AS expression was also observed but was significantly lower. Ampicillin stimulated C29 switch-off pheromone expression in recipient cells, which in turn switched off AS expression in donor cells. We observed that although *prgB* was switched off after 0.5× stimulation in C29, the supernatants induced expression in certain 1390R strains. In conclusion, ampicillin was able to modulate pheromone expression in free plasmid clones which, in turn, modulated AS expression in plasmid donor cells. The fact that PrgB gene expression was switched off after the ampicillin stimulus at 0.5× MIC, whereas AS proteins were present on the surface of the bacteria, suggested that a mechanism of rescue associated with mechanism pheromone sensing may be involved.

## 1. Introduction

*Enterococcus faecalis*, a gram-positive coccus, is part of the normal microbiota but is clinically relevant because of its high ability to develop antimicrobial resistance encoded on plasmids [1,2]. In enterococci, the transfer of antimicrobial resistance-associated plasmids is mediated by pheromone sensing [3]. In most cases, transfer involves the participation of small hydrophobic peptides called pheromones (Phe). According to the model of the plasmid pCF10, the phe encoded by *ccfA* is constitutively produced from plasmid-free enterococcal strains and released to the medium upon maturation [1]. Upon release, phe interacts with factors encoded by pCF10 to induce conjugative transfer [4]. Aggregation substances (AS) participate in the mating of donor and plasmid recipient cells during bacterial conjugation, enabling plasmid transfer and acquisition of antimicrobial resistance [5]. Selective pressure is involved in the increase in nonplasmid-acquired resistance [6]. The emergence of new mutations that affect the action of antibiotics and the selection of resistance at either lethal or nonlethal concentrations have been systematically analyzed [7,8]. The pheromone response associated with an increase in clumping of *E. faecalis* upon peptide sex cAD1 expression under the selective pressure of tetracycline, erythromycin or chloramphenicol has been reported [9]. It is known that in the horizontal transference of plasmids, the mechanisms of pheromone sensing are the form in which the genus *Enterococcus* acquires plasmids that can harbor resistance genes. Several of them are well characterized and can even carry genomic islands from the receptor [10].

In this context, the role of stimuli with antibiotics on the response of pheromones and responsive plasmids to pheromones and aggregation substances of donor cells has been poorly studied. This should be analyzed because it opens an overview of the mechanisms of transference and emergence of resistance, as well as of dissemination of markers of resistance induced by antibiotics from resistant clones to sensitive clones in *E. faecalis*. The purpose of this study was to analyze and evaluate the effect of supernatants rich in pheromones produced under stimulation with subinhibitory concentrations of ampicillin from a plasmid-free clone stimulated with different sub-inhibitory concentrations of ampicillin on the expression of AS in a plasmid-donor clone in which both strains had the same *E. faecalis* clinical strain origin.

## 2. Materials and Methods

### 2.1. Bacterial Strains and Growth Conditions

A clinical isolate of *E. faecalis* (1390) was tested for penicillin, gentamicin, and streptomycin resistance as well as high-level streptomycin resistance (HLSR) and high-level gentamicin resistance (HLGR) according to Clinical and Laboratory Standards Institute (CLSI) guidelines [11]. This strain was selected from a susceptibility profile of 90 clinical strains by the Instituto Nacional de Cardiología in Mexico City and based on a plasmid analysis performed according to a previously published report [12] as well as the phenotype and genotype of HLAR and *prg* B (Appendix A). The strain was subjected to selective pressure with ampicillin (Sigma-Aldrich, Saint Louis, Missouri, USA) continuously at subinhibitory concentrations (0.25×, 0.5× and 0.75×) as previously described [13]. The recovered clones were thereafter stimulated continuously with higher concentrations of ampicillin minimum inhibitory concentration (MIC) (1×, 2×, 4×, 8×, 16×, 32×, 64×, and 128×); afterward, the resistant clones were analyzed based on a plasmid profile. The resistant clone with plasmid was named 1390R and considered a donor clone and able to express aggregation substances (AS). The 1390R strain was subjected to treatment with ascorbic acid (2, 5, 10, and 15 mM) to induce plasmid loss as previously reported for *Staphylococcus* [14]. The plasmid profile of the resulting clones was tested, and a resistance analysis was conducted for ampicillin as well as HLGR and HLSR. The recovered clone that was plasmid-free of plasmids and sensitive to the tested antibiotics, was named C29 and considered a receptor and pheromone producer. Susceptibility was determined by an analysis of viability. Briefly, the strains were grown for 6 h at 37 °C, and the pellet was recovered by centrifugation and adjusted to 0.5 McFarland standard (1.4 × 10^8^ bacterial/mL). The MIC of ampicillin of each strain was tested following the CLSI criteria. The tubes were grown for 4 h as previously described, and the bacterial suspension was diluted to 10,000 bacteria/mL in PBS 1× (pH 7.4) and EDTA 0.05 M. Rhodamine 123 (Sigma-Aldrich) was added to a concentration of 2 µg mL^−1^, and propidium iodide (Sigma-Aldrich ) was added to a concentration of 10 µg mL^−1^. Each dye was added to the bacterial suspensions. To establish the kinetics of survival in each strain, the strains were analyzed by flow cytometry in a FACSCalibur device with CellQuest software (Beckton Dickinson, Becton Drive Franklin Lakes, NJ, USA).

### 2.2. Affinity of Penicillin-Binding Proteins (PBPs)

The affinity of penicillin-binding proteins (PBPs) to penicillin was analyzed using a conjugate of (+)6-aminopenicillanic acid (Flu6-APA) (Sigma-Aldrich) and N-hydroxysuccinimide-activated-fluorescein (Sigma-Aldrich). The assay was performed according to a previously published procedure [15]. Each strain was grown for 6 h at 37 °C and centrifuged, and the pellet was washed twice with PBS 1× (pH 7.4) buffer. The cells of the pellet were permeabilized with saponin and then added to the conjugate to reach an estimated concentration of 10 µM. The cells were incubated at 37 °C for 30 min and washed twice with 1× PBS (pH 7.4). Afterward, the cells were analyzed by flow cytometry in a FACSCalibur device with the software CellQuest (Beckton-Dickinson); the percentage of gates and mean fluorescence intensity were obtained to determine the percentage of cells marked with the conjugate penicillin-fluorescein.

### 2.3. Analysis of HLSR

To determine the emergence of HLSR clones, a culture in broth with 64× MIC was recovered and analyzed using HLSR disks following the Kirby-Bauer method (BBL^TM^ Beckton Dickinson) [11]. Additionally, DNA was extracted from this culture with Gentra Puregene (Qiagen, Turnberry Lane Suite 200, Valencia, CA, USA) according to the manufacturer’s instructions. Purified DNA and plasmids were used as the template for PCR using primers targeting the HLSR gene *ant(6)-I*. The 577-bp product was obtained following a previously reported procedure [15].

### 2.4. Isolation of Cell Supernatants under Selective Pressure of Ampicillin

The plasmid-free C29 strain was grown in BHI broth without antibiotics and supplemented with ampicillin at 0.25×, 0.5×, and 0.75× MIC (Sigma‒Aldrich). The culture was incubated for 18 h at 37 °C under shaking at 200 rpm, followed by centrifugation at 10,000× *g*. The supernatant was filtered with a 0.22-µm membrane followed by ultrafiltration in an Amicon membrane^®^ (PM30 polyethersulfone, Merck Millipore, Burlington, MA, USA) and sterilization at 121 °C/20 min as previously reported [1,3].

### 2.5. Clumping Assay

To determine the influence of supernatants produced under ampicillin stimulation on the aggregation of 1390R and C29 strains, a clumping assay with the donor (1390R) and the recipient (C29) strains was performed as previously reported [15,16]. Assays were performed with the following modifications: the 1390R strain was grown for 18 h at 37 °C, recovered and washed with PBS 1× as was previously reported. To obtains early expression of AS on the donor before the interaction, we incubated the donor-receptor interaction for 2 h at 37 °C in the presence of media enriched with pheromones from the C29 strain.

After this time, the donor strain interacted with the recipient C29 in a U-bottom plate, and aggregation was monitored in a plate reader at a controlled temperature (GENios Tecan, Seestrasse Männedorf, Switzerland). The plate temperature was maintained at 37 °C, and readings were performed every two min at 630 nm for 90 min. The decrease in absorbance was interpreted as bacterial donor-receptor aggregation.

### 2.6. Expression of prgB and Iad1 in the Stimulated 1390R and C29 Strains

RNA was extracted from a culture of 1 × 10^6^ cells of the 1390R strain stimulated with each supernatant (0, 0.25×, 0.5×, and 0.75× MIC) using the RiboPure™-Bacteria Kit (Thermo Fisher Scientific, Waltham, Massachusetts, USA) according to the manufacturer’s instructions. cDNA was generated with a Gene Amp RNA Kit, a High-Capacity cDNA Reverse Transcription Kit (Applied Biosystems Carlsbad, CA, USA), and 2 µg of extracted RNA. Once the cDNA was synthesized, a PCR assay of *prgB* was performed according to a previously published procedure [16], and a product of 457 bp was observed upon agarose gel electrophoresis. In addition, a real-time analysis was performed with 5 µL of cDNA and a SYBR^®^ Green PCR Master Mix Kit (Applied Biosystems) following the manufacturer’s instructions. The analysis was run on an ABI Prism 7000 system (Applied Biosystems), and the relative expression was analyzed using the nonstimulated cells as the calibrator and each stimulus as the test. To determine the degree of increase in the stimulus with antibiotic, the test/calibrator ratio was calculated. We applied the following equation: ratio = 2 (Ct calibrator-Ct test). A pair of primers (sense 5′ATAAGAGGAGAGCTATTAGAATG-3′ and antisense 5′-GGGGAATATACATGAAC-3′) was designed to amplify the *iad1* gene, which inhibits pheromone expression, based on the sequence M62888.1. The *iad1* gene encodes the sex pheromone inhibitor (iAD1) on the plasmid pAD1, which is expressed in response to the pheromone, and the fragment length is 417 bp.

### 2.7. Analysis of AS Expression in the Ampicillin-Stimulated 1390R Strain

Aggregation was analyzed by confocal microscopy, flow cytometry, and ELISA as follows. (a) Confocal microscopy: when the cultures of the stimulated 1390R strain reached 0.5 McFarland standard, an aliquot was obtained, washed, and resuspended in PBS 1× (pH 7.4). An aliquot was deposited on a glass slide coated with polylysine and incubated overnight at 4 °C in a humid chamber. Thereafter, the glass slides were washed twice with wash buffer (Dako, Carpinteria, CA, USA), and an anti-AS antibody diluted at a ratio of 1:50 (from Dr. Gary Dunny, Minnesota University) was added, incubated for 30 min at room temperature, and then washed twice with wash buffer (Dako). A secondary anti-mouse IgG (gamma) antibody, which was an F(ab’)2 fragment, human serum adsorbed and FITC labeled (Dako), was added at a dilution of 1:300. The glass slides were washed twice and analyzed using a confocal microscope (Carl Zeiss-Strasse 22 Oberkochen, 73447 Germany) with the software Axio Vision LE version SE64 rel. 4.9.1 (Carl Zeiss). (b) Flow cytometry: Cells were added to a tube at 100,000 cells/mL, and the previously described procedure was followed, except that anti-AS was diluted at a ratio of 1:600 and the anti-mouse antibody was diluted at a ratio of 1:100. Once labeled, each tube of cells was read in a FACSCalibur flow cytometer (Becton Dickinson) and analyzed using CellQuest pro software v 5.2.1. (c) ELISA. One milliliter of 1 × 10^6^ cells/mL was recovered, centrifuged, washed twice with PBS 1× and resuspended in 20 µL of PBS 1×. Each supernatant of stimulated 1390R was placed in a 96-well microplate and shaken for 5 min at low velocity; then, the plate was incubated overnight at 60 °C. The wells were washed with PBS 1×, and a rabbit anti-AS polyclonal antibody was added at a dilution of 1:100. The plate was incubated for 3 h at room temperature, the wells were washed 6 times with PBS-Tween 20, and anti-rabbit antibody labeled with phosphatase was added and incubated for 30 min. Development was performed with paranitrophenol phosphate and stopped with sulfuric acid (0.5 M), and the absorbance at 405 nm was read in an ELISA GENios TECAN plate reader.

### 2.8. Statistical Analysis

The experiments were performed in triplicate, and the statistical analysis was performed with GraphPad Prism 8 software (La Jolla Cal. the USA), and significance was considered with a *p* < 0.05 using ANOVA of one and two ways. Comparative analysis between groups was done by the use of Tukey’s multiple comparison tests, whereas a comparative analysis between the control vs. experimental groups was done with Dunnett’s multiple comparisons test, and Sidak’s test multiple comparison tests was used for the longitudinal integral analysis.

## 3. Results

### 3.1. Plasmid-Free E. faecalis Obtained β-Lactam Resistance Prior to Ampicillin Treatment

The plasmid profile of *E. faecalis* strain 1390 showed the absence of plasmids and an MIC of 1 µg mL^−1^ for ampicillin, HLAR sensitivity for gentamicin and streptomycin, but carried aac(6′)-aph(2′′)-I and ant(6′)-I as well as *Prg B*. This strain was named 1390S (Appendix A, line 1). The MIC of 1 µg mL^−1^ for ampicillin was considered 1×. The 1390S strain was stimulated with 0.25× to 1× (from 0.5 µg mL^−1^ to 2 µg mL^−1^) MIC, and the kinetics of viability were measured by flow cytometry (Appendix A). This method was used to increase the sensitivity of detection. The bacterial viability decreased by 20% upon exposure to 0.25× MIC, 75% at 0.5× MIC, 86% at 0.75× MIC, and 99% at 1× MIC (Appendix A). The 1% 1× MIC-viable strain was recovered and again stimulated with high ampicillin concentrations up to 64×, equivalent to 128 µg mL^−1^ ampicillin. The resistant clones were recovered to 32× (equivalent to 64 µg ml^−1^ (Figure 1A), and the MIC was performed. A resistant clone was isolated (Appendix A), and a growth curve was performed at 1× and 2× MIC. The results did not show inhibition, suggesting the selection of a resistant strain (Figure 1B). These strains isolated to 64× MIC were defined as 1390R. The ampicillin affinity was measured using a conjugate of penicillin and fluorescein. In the 1390R strain recovered from stimulation with 64× MIC, a 100% loss in penicillin fixation was observed, likely indicating positive selection (Figure 2A–F).

### 3.2. Ampicillin Resistance in the E. faecalis Strain 1390R Is Associated with an ~47-kb Plasmid

The plasmid profile of the 1390R strain showed the presence of an ~47-kb plasmid. This strain was cured of the plasmid using a double ascorbic acid treatment at 10 mM, and 5% of the clones were free of the plasmids. From these clones, a clone was selected and named C29. The C29 strain showed an MIC of 1 µg mL^−1^ for ampicillin, similar to the 1390S strain (Appendix A). The 1390R strain 64× also showed an HLSR phenotype; this was confirmed by the detection of the gene *ant(6)-I*, which is responsible for streptomycin resistance. The *ant(6)-I* gene was absent in the C29 strain, which was confirmed by the absence of the plasmid in the plasmid profile and the loss of the HLSR phenotype. In addition, expression of *iad1* (pheromone, plasmid-encoded peptide) was observed in the 1390R strain but not in the C29 strain, whereas the *prgB* gene encoding the AS molecule (Asc10) expressed by pheromone-induced cells carrying plasmid pCF10 was not identified in C29 (Appendix A). Primers for *iad1* were designed based on the sequence downloaded from the GeneBank under accession number M62881 (Appendix A), and amplification was observed in 1390S and 1390R, suggesting that the wild-type strain, as well as the resistant clone isolated after the ampicillin stimulus, both express the pheromone inhibitor, with more intensity in 1390R. This is because the density of the population with plasmid encoding *iad1* in the resistant clone (1390R) is higher, once resistant clones have been isolated, whereas 1390S is a mix between resistant and sensitive clones (Appendix A). The *prgB* gene encodes aggregation substance in the plasmid (Appendix A), and *ant(6)-I*, which is responsible for HLSR (Appendix A), was found in 1390S and 1390R but not in C29.

### 3.3. Effect of the Supernatant from Ampicillin-Stimulated E. faecalis Strain C29 on Clumping

Enrichment pheromone medium was obtained from the C29 strain exposed to different concentrations of ampicillin. Clumping was observed at 6 min (Figure 3A) for all concentrations, and maximal clumping was observed with the pheromone medium produced at 0.25× MIC (Figure 3B). The supernatant of C29, both with and without 0.25× MIC, only stimulated the 1390R strains to express *prgB*; however, the supernatant with 0.5×, 0.75×, and 1× could not stimulate the expression of *prgB* (Appendix A). The difference in relative expression (RE) between 1390R stimulated without ampicillin and 1390R stimulated with 0.25× was only 0.5-fold, and no expression was confirmed for 1390R stimulated with 0.5× and 0.75× MIC. When *prgB* expression was semiquantified, no difference was found between the different stimuli and the control (*E. faecalis* ATCC 51299) and 1390R (Appendix A). We investigated the effect of conditioned supernatants from 0.25×-to 0.75×-stimulated C29 on the 1390R strain (the 1× MIC concentration was omitted to avoid the total inhibition associated with the antibiotic on C29), and the switching off of expression was confirmed. Confocal microscopy revealed that the greatest increase in AS expression occurred for the 1390R strain with the supernatants from stimulated C29 (Figure 4). When the 1390R strain was exposed to supernatants with 0.5× and 0.75× MIC, expression of AS was also observed; however, AS expression was significantly lower when the 1390R strain was exposed to the supernatant from C29 stimulated with 0.75× MIC (Figure 4). The ELISA results confirmed these data (Figure 5A) When the 1390R strain was stimulated with supernatants of ampicillin-stimulated C29 and analyzed by flow cytometry, AS expression and the number of events were maximal at 0.25× MIC. However, when the 1390R strain was exposed to the supernatant from 0.5× MIC-stimulated C29, a dramatic decrease in events was observed. However, the fluorescence intensity was maintained (Figure 5B).

## 4. Discussion

Pheromone sensing is the mechanism normally used by *Enterococcus* for plasmid transfer from a donor cell to a receptor cell [16,17]. However, the process of signaling for transfer of a pheromone-specific plasmid from a donor to receptor cell is complex [18]. The acquisition of different pheromone-dependent plasmids can enable the acquisition of several virulence factors in the form of cell receptors: antimicrobial resistance markers or factors that promote cytolysis as well as increased adherence to eukaryotic cells [19,20]. Some evidence suggests that external stimuli can change donor-receptor aggregation in response to antibiotics or external peptides [9,21,22]. In addition, when ampicillin interacts with sensitive enterococci, the antibiotic reacts with its target, which is PBP [23]. This event triggers an irreversible binding of penicillin to PBP [11], which can lead to cellular stress with biochemical and biophysical changes. Based on these phenomena, we analyzed the role of subinhibitory ampicillin stimuli on the pheromone-sensing mechanism of sensitive enterococci by using a donor and receptor from the same clinical strain (1390). The donor (1390R) was obtained by a subinhibitory stimulus, and the receptor was subsequently obtained by curing the 1390R strain to obtain a plasmid-free clone (C29). A clone of the 1390R strain highly resistant to ampicillin was recovered after a subinhibitory stimulus of ampicillin. Previous reports described the association of low-affinity PBPs, which are encoded on plasmids and carried on mobile genetic elements such as transposons [11]. We observed the presence of plasmids in clones that were selected under selective pressure. The kinetics of activity were analyzed in the highly resistant clone (1390R strain). The most likely reason for resistance was the association with overproduction of low-affinity PBPs [4,16], and our results confirmed that the continuous stimulus from subinhibitory to high concentrations of ampicillin induced the selection of clones with plasmids encoding resistance to ampicillin. In addition, we observed the emergence of HLSR associated with *ant(6)-I* in the 1390R strains recovered after the ampicillin stimulus, suggesting that during clone selection induced by ampicillin, cross-selection of other antibiotic markers was also possible, although we only detected HLSR and HLGR. However, we could not determine whether more markers for other antibiotics were present in the 1390R strain. Thus, our results showed a possible coselection and thus coresistance in the clones selected after the stimulus with ampicillin [5]. High-level aminoglycoside resistance (HLAR) expression has been observed in lactobacilli exposed to gentamicin [24]. These observations and our findings are evidence that subinhibitory concentrations of antibiotics such as ampicillin can trigger HLAR expression.

To obtain a plasmid-free strain, we used a method previously tested for *Staphylococcus* and an ascorbic acid concentration of 1 mM [13]. For our 1390R strain, we used an initial treatment with 10 mM ascorbic acid, but we could not increase the concentration because ascorbic acid was 100% toxic to our strain. The bottom cells from a 24 h culture was newly treated with 10 mM ascorbic acid. In this second step, the plasmid-free clone C29 was recovered, and testing was performed for the markers ampicillin, HLGR and *iad1*, which encode the inhibitor pheromone pAD1 hemolysin/bacteriocin [25]. We expected that the loss of *prgB* expression influenced by the AS encoded by plasmid pCF10 would not be present in the C29 strain. Our results indicated that with the use of a high concentration of ascorbic acid (10 mM), several markers (resistance and virulence factors) encoded by plasmids in *E. faecalis* were eliminated in two steps. Wild-type *E. faecalis* can carry multiple plasmids [14]. Thus, the ability to cure a strain with multiple plasmids or plasmids with multiple markers of virulence is an important finding during the search for factors to control the emergence of antibiotic resistance. *Enterococcus* has several plasmids and multiple copies of these plasmids, which explains the difficulty of curing *Enterococcus*; however, the curing of resistant strains indicates the feasibility of reducing possible markers of resistance in multidrug resistant strains that are known to coexist with commensal strains [26].

The absence of a specific plasmid is known to trigger the expression of a specific pheromone [1,27]. We tested whether the ampicillin stimulus affected the mating of the donor and receptor. The C29 and 1390R strains were induced to mate in the presence of supernatants from C29 that were produced without ampicillin and with different subinhibitory concentrations of ampicillin. The best mating of the receptor and donor was observed when the conditioned medium containing pheromone was produced in the absence of the antibiotic and 0.25× MIC or when these conditioned media were diluted from 1:2 to 1:64, which was the maximum dilution tested. We hypothesize that when the concentration of antibiotic is increased to 0.25×, the bacterial population begins to decrease, which affects pheromone production and thus the effect of mating. However, such a hypothesis cannot be true because the effect of mating induced by the pheromone medium was still observed at concentrations up to 0.5×, suggesting the presence of the pheromone. Additionally, as noted in previous work, when the pheromone medium from the stimulus without antibiotic was diluted, a better effect of mating was observed. This finding suggests that the diluted pheromone medium can induce better mating between donor and receptor compared with pheromone medium produced in the absence of antibiotic.

The best mating was observed when such medium was produced in the absence of ampicillin as well as with 0.25× MIC. When the antibiotic was increased up to 0.5× MIC, mating was decreased, suggesting an effect on aggregation. Analysis of the expression of the *prgB* gene, which encodes AS, showed that expression occurred when the 1390R strain was stimulated with the pheromone medium without antibiotic and with 0.25× MIC, with a 0.5-fold difference in expression between them, whereas the expression of the *prgB* gene was turned off when stimulated with pheromone medium with 0.5× and 0.75× MIC. These results indicate that the pheromone medium obtained with a low stimulus of ampicillin (0.25×) upregulated the surface expression of AS on donor cells, whereas when the antibiotic concentration was increased in medium containing the C29 strain (0.5× and 0.75×), the expression of AS by the donor was switched off. We tested the expression of AS on the 1390R strain using a polyclonal and monoclonal anti-AS antibody (from Dr. Gary Dunny) and measured the expression by confocal microscopy, ELISA, and flow cytometry. Although immunoblotting is the best method for this type of analysis, we used a combination of methods that also allowed us to analyze live bacterial cells. A considerable increase in AS expression was observed in all experiments when 1390R was stimulated with pheromone medium produced from C29 with 0.25× MIC; at higher concentrations of ampicillin, a rapid decrease in AS expression was observed to be below even the level induced by pheromone medium obtained without ampicillin. However, when the 1390R strain stimulated under different conditions was analyzed by flow cytometry, the number of detected cells expressing AS on the surface differed, as measured by intensity (events and mean fluorescence intensity (MFI). However, when the cells stimulated with 0.25× MIC were analyzed, the number of cells expressing AS and the intensity of AS on the cell surface were consistent. When the stimulus was increased to 0.5× and 0.75×, a rapid decrease was observed in the number of cells (events) expressing AS, although the intensity of those cells that expressed AS was high. These results suggest that under stimulation with the pheromone medium, some cells of the 1390R strain increased the surface expression of AS considerably, whereas other clones maintained the switched-off status of the system. We believe that some clones respond intensely to the pheromone medium produced in the presence of an antibiotic at concentrations above 0.5× MIC, while others do not respond. In this context, we hypothesize that certain clones of the C29 strain were capable of responding while others did not respond to the stimulus of the supernatant with pheromone. In this work, we did not evaluate whether the effect of ampicillin on the C29 strain could modulate pheromone expression because previous studies showed that the receptor cells have their own mechanism of control to avoid overproduction, which can lead to toxicity [26,28]. Furthermore, we can presume that during the first stimulus (0.25×), the antibiotics upregulated pheromone production. However, when the antibiotic concentration was increased to 0.5× MIC, pheromone production could not be increased due to the toxic effect. Another alternative is associated with bacterial fitness [8]. In the first stimulus with ampicillin, C29 was able to trigger pheromone secretion to attract the bacterial plasmid donor. However, when the antibiotic concentration was increased, the bacterial cell utilized other methods of maintaining viability and bacterial fitness in the presence of the antibiotic [29]. In addition, AS expression depends on a type IV secretion system [12] that involves several mechanisms of activation and repression [30]. It is likely that an additional mechanism participates in the induction and repression of AS expression, perhaps these are substances present in the supernatant from the C29 strain stimulated with 0.5× and 0.75× and generated during bacterial stress in the presence of ampicillin. In addition, AS is considered a virulence factor because it mediates adhesion between enterococci and eukaryotic cells [31,32]. The different supernatants had similar effects on the adhesion of 1390R, and adhesion was reduced after stimulation of 1390R with the supernatant from C29 at 0.5× MIC. In conclusion, our results suggest that ampicillin stimulation of the C29 strain, free of plasmids, modulated the expression of pheromones, increasing it at low concentrations (0.25× of MIC) and decreasing it at high concentrations (up to 0.5× MIC of ampicillin), which in turn modulated the expression of AS in the donor cells. We observed that although PrgB expression was switched off after antibiotic stimulus at 0.5× MIC in the C29 strain, the high-intensity supernatants were able to induce expression in some 1390R cells. This finding suggests the existence of a mechanism to rescue pheromone expression in the C29 strain free of plasmid. Our results offer an overview of how antibiotics such as ampicillin can modulate the response of pheromones in receptor cells, and, in turn, modulate the aggregation expression substances. This mechanism may be associated with the horizontal transference and dissemination of resistance markers from resistant clones to sensitive clones of *E. faecalis*.

## Figures and Tables

**Figure 1 microorganisms-10-02294-f001:**
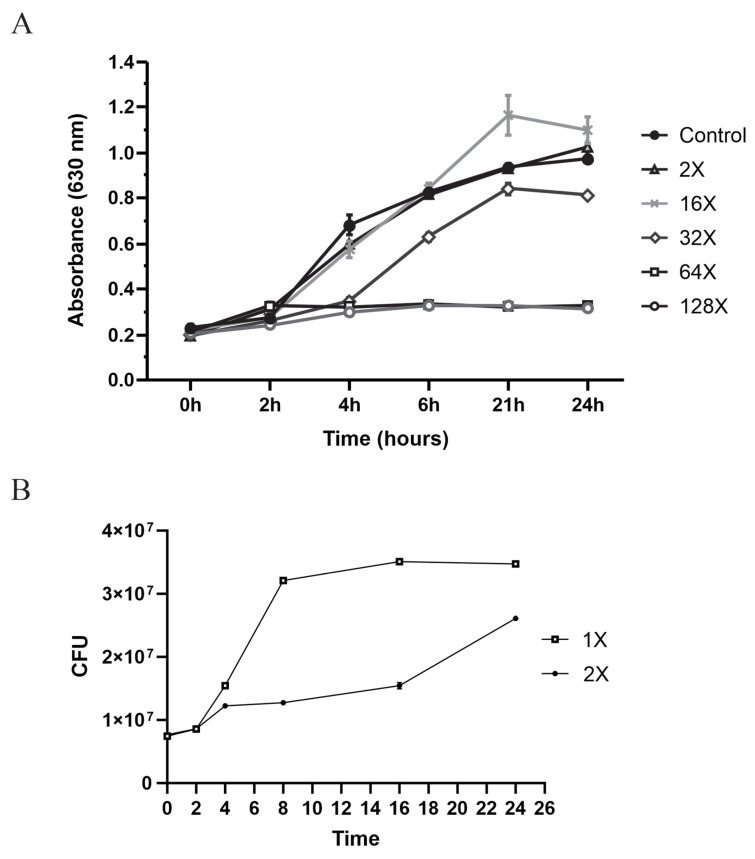
Analysis of the viability of the 1390S strain after continuous stimulation with high concentrations of ampicillin at 16, 32, 64, and 128 µg/mL. (**A**) Growth curves in presence of selective pressure (*p* < 0.0001). (**B**) Recovery of resistant strains from the high-stimulus conditions and growth at 1× and 2× MIC (2 and 4 µg/mL) (*p* < 0.01).

**Figure 2 microorganisms-10-02294-f002:**
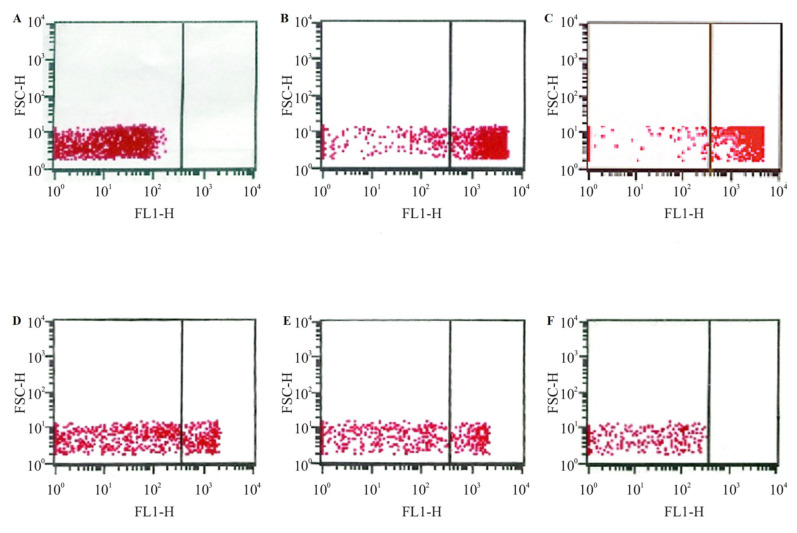
Activity of penicillin-binding protein (PBP) in clones recovered at 64× MIC measured by flow cytometry. (**A**) Control 1390S without fluorescent ampicillin, (**B**) control 1390S with fluorescent ampicillin, (**C**) control strain 29212, control of ampicillin-sensitive *E. faecalis*, (**D**) strain 1390S stimulated with 0.25× MIC, (**E**) strain 1390S stimulated with 0.5× MIC, and (**F**) strain 1390R stimulated with 64× MIC. The resulting bacterial clone (1390R) had a total loss of ability to bind PBP or penicillin.

**Figure 3 microorganisms-10-02294-f003:**
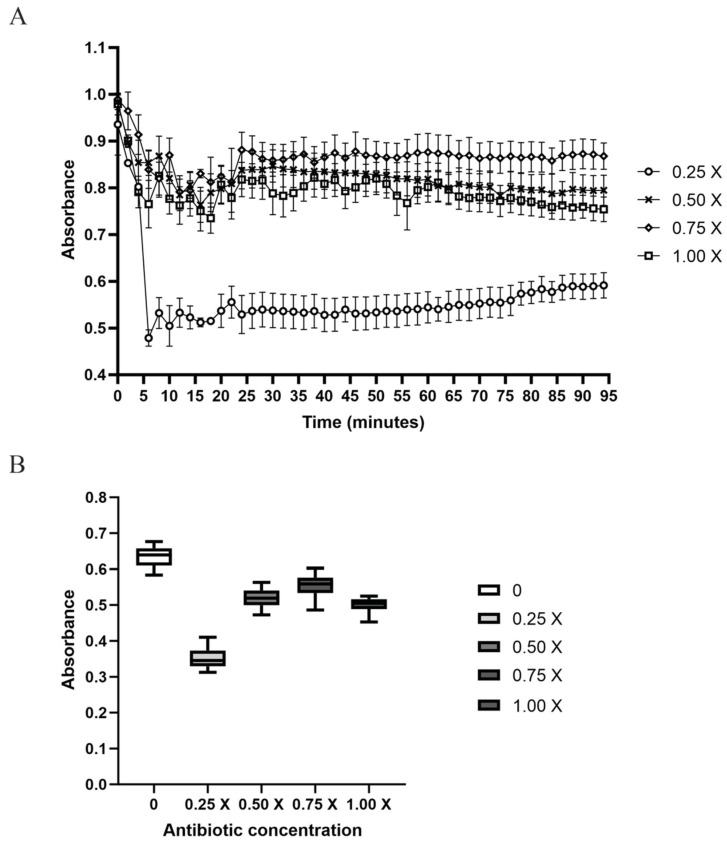
Effect of the supernatants from C29 strain on the clumping of C29 and 1390R. (**A**) Spectrophotometric analysis of clumping in the presence of pheromone medium produced with different concentrations of ampicillin (0.25×, 0.5×, 0.75×, and 1× MIC) (*p* < 0.0001). (**B**) Clumping in the presence of different concentrations of antibiotic. Clumping activity was highest at an antibiotic concentration of 0.25× MIC (*p* < 0.0001).

**Figure 4 microorganisms-10-02294-f004:**
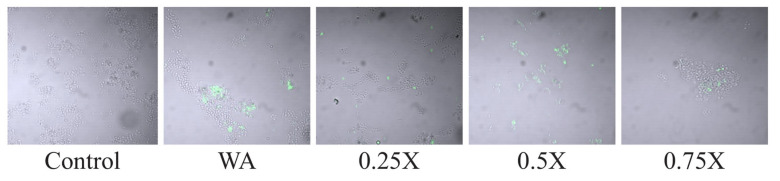
Effect of the supernatants from C29 obtained without stimulus or with ampicillin stimulus at 0.25×, 0.5×, and 0.75× MIC on 1390R. Confocal microscopy analysis was performed. The 1390R was stimulated with the supernatant of C29 produced in the absence of ampicillin and with 0.25×, 0.5×, and 0.75× MIC.

**Figure 5 microorganisms-10-02294-f005:**
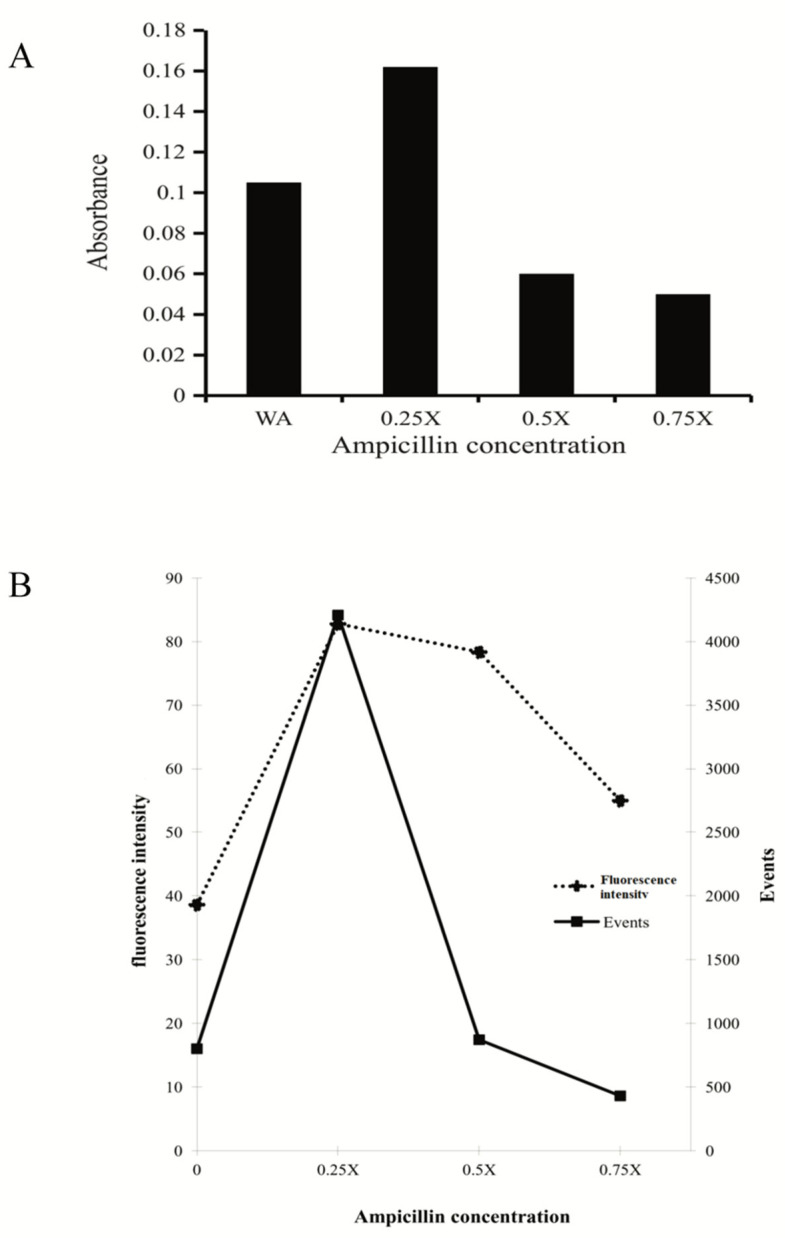
Expression of aggregated substance by the 1390R strain upon exposure to different stimuli. (**A**) ELISA for the detection of AS in the presence of different concentrations of antibiotic (comparative analysis between groups: WA vs. 0.25×: *p* < 0.0022, WA vs. 0.75: *p* < 0.0165, 0.25× vs. 0.5×: *p* < 0.0002, 0.25× vs. 0.75 *p* < 0.0001) (comparative analysis of the experimental groups vs the control: WA vs. 0.25× P= 0.0013, WA vs. 0.75×” *p* = 0.0099). (**B**) Analysis of AS expression on the cell surface by flow cytometry. The mean fluorescence intensity represents the total detected fluorescence, whereas events represent the count of fluorescent cells (one way ANOVA *p* < 0.0001. Tukey’s multiple comparison test: 0 vs. 0.25× *p* <0.0001, 0 vs. 0.50× *p* < 0.0001, 0 vs. 0.75× *p* < 0.0028, 0.25 × vs. 0.50 × *p* = 0.4825, nonsignificant, 0.25× vs. 0.75× *p* < 0.0016, and 0.50 × vs. 0.75 × *p* < 0.0099, Dunnett’s multiple comparisons test 0 vs. 0.25× *p* < 0.0001, 0 vs. 0.50× *p* < 0.0001, 0 vs. 0.75× *p* < 0.0016).

## Data Availability

All relevant data have been included in this manuscript.

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
