# Peer review of "Pheromone Activity after Stimulation with Ampicillin in a Plasmid-Free Enterococcus faecalis Strain"

_microorganisms, 2022, doi:10.3390/microorganisms10112294_

Round 1
Reviewer 1 Report
Pheromone-perponsive plasmids in E. faecalis are the most extensively characterized plasmids paticularly among Gram-positive pathogenic microorganisms includig Enterococcus. This study provides additional information based on pheromones activity from stimulation with ampicillin in a plasmid free E. faecalis strain. Plasmid free E. faecalis excrete peptides which specifically induce a mating response in strains harboring certain conjugative plasmids. The response is characterized by the synthesis of a fuzzy surface material, visible by electron microscopy (EM). Therefore the use of EM in this study may improve the quality of the manuscript and allow to make clear interpretation of the results.
Author Response
We appreciate the observation and agree with the reviewer that electron microscopy should improve the quality of results. However, we could not have it available due to a lack of technical experience to prepare our samples; alternatively, we resorted to confocal microscopy and flow cytometry, which showed considerable sensitivity. Additionally, the flow cytometry allowed made a semi-quantification of the fluorescence.

Reviewer 2 Report
The work titled Pheromones activity from stimulation with ampicillin in a plasmid-free Enterococcus faecalis strain need mote revision. The authors failed to clarify the aim of the work in the introduction and doe not clarify the research question.
in the abstract the authors need to wrap up the work
in introduction, no data about other products activities
in method, analysis of tests need to be in details
Author Response
In the abstract: A conclusion was included.
In introduction: This was complemented even with new information
In methodology. Some confused points were cleared and improved
Results. The supplemental figure 3A was substituted; we believe that the new figure 3A offers more results to have a more robust conclusion
The conclusion was improved as well.

Round 2
Reviewer 2 Report
It is ok